# OpenReview forum: "Deep-Cover Agents: Long-Horizon Prompt Injections on Production LLM Systems"
_ICLR.cc/2026/Conference — Submitted to ICLR 2026_

### Official Review · Reviewer_CGsz · 2025-10-28

**Soundness:** 2
**Presentation:** 2
**Contribution:** 2
**Rating:** 2
**Confidence:** 5

**Summary:**

This paper presents a framework for analyzing prompt injection attacks against tool-using LLM agents that process untrusted data. The authors identify understudied attack modalities and examine how attack performance varies based on injection method and token budget, yielding practical insights for both attackers and defenders.

A key finding is the discovery of "delayed" prompt injection attacks, where compromised agents can behave normally for over 50 conversation turns before executing malicious actions—a temporal dimension largely unexplored in prior work. The research is validated through successful sandboxed attacks against real-world deployment systems including Claude Code and Gemini-CLI, which also reveal previously undocumented emergent behaviors in these models' responses to injection attempts.

**Strengths:**

1.The paper makes an important and understudied contribution by investigating how prompt-injected agents can exhibit delayed malicious behavior. This temporal dimension has been largely overlooked in prior work and has implications for detecting and mitigating prompt injection attacks in real-world deployments, as it suggests that simple immediate monitoring may be insufficient.
2. Prompt injection is a critical security concern for tool-integrated agents, and this paper identifies understudied attack modalities and examine how attack performance varies based on injection method and token budget, yielding practical insights for both attackers and defenders.

**Weaknesses:**

I think the main problem is the limited novelty in threat model and the contribution is limited as well.

1. The work appears to focus on direct prompt injection where attackers can modify system prompts, which is impractical. Real-world scenarios involve indirect prompt injections through untrusted data (emails, documents, web pages). The threat model may be too strong and less relevant than existing work on indirect attacks.
2. The delayed/time-based attacks (behaving benignly for 50+ turns) seem more like an interesting observation than a critical security concern. If we cannot effectively defend against immediate prompt injections, delayed attacks are a secondary issue. The practical significance is unclear.
3. The evaluation relies solely on ASR, while comprehensive benchmarks should also measure utility/benign task performance. This is critical for assessing the practical trade-offs of defenses and the true impact of attacks on system usability, for example agentdojo.

4. Figure 1 is not referenced or explained in the text. Figure 3 (top left) lacks a legend making it difficult to interpret. Additionally, there are formatting issues with incorrect quotation mark usage (line 38 and 43).

**Questions:**

no

---

> ### Author Response · Authors · 2025-12-03
>
> We thank the reviewer for their detailed feedback. We would like to offer the following responses to points raised.
>
> > The work appears to focus on direct prompt injection where attackers can modify
> system prompts, which is impractical. Real-world scenarios involve indirect prompt
> injections through untrusted data (emails, documents, web pages). The threat model
> may be too strong and less relevant than existing work on indirect attacks.​
>
> We want to emphasize that, for the current coding agents that are the most popular in the world right now, the system prompt can be directly modified. This is a real-world threat model – and it is of particular relevance given that previous work on indirect attacks, by focusing on the Tool vector, may have made developers fixate on that attack vector at the expense of other options (see Figure 3: top left).
>
> > The delayed/time-based attacks (behaving benignly for 50+ turns) seem more like an
> interesting observation than a critical security concern. If we cannot effectively defend
> against immediate prompt injections, delayed attacks are a secondary issue. The
> practical significance is unclear.​
>
> We agree that prompt injections in general are an unsolved security concern. However, work towards solving prompt injection could be ineffective if it does not effective cover the full range of possible threat models, as narrower interventions (e.g. reviewing a single turn of conversation at a time) may be insufficient to identify all attacks. With this work, we intend to encourage the field to consider a broader range of threat models, so that solutions to immediate prompt injection attacks are more likely to be built with an eye towards generalization.
>
> Importantly, we think this covers an extremely practical threat model: A user supervising their coding agents monitors its actions for a few turns after it accesses untrusted data, but several hours later, has forgotten their suspicion and trusts the still-correctly functioning agent, which has secretly gained a sleeper-agent like behavior.
>
> > The evaluation relies solely on ASR, while comprehensive benchmarks should also
> measure utility/benign task performance. This is critical for assessing the practical
> trade-offs of defenses and the true impact of attacks on system usability, for example
> agentdojo.​
>
> We excluded benign task performance as, in all results shown in this paper, benign task performance was ~100% (except where the model refused to do the benign task by informing the user about the covert task it was instructed to perform, which we still consider a task success). We think this is reasonably reflective of real-world deployments, as many LLMs will be used for rote or routine tasks that are not at the frontiers of SWE performance.
>
> > Figure 1 is not referenced or explained in the text. Figure 3 (top left) lacks a legend
> making it difficult to interpret. Additionally, there are formatting issues with incorrect
> quotation mark usage (line 38 and 43).​
>
> We thank the reviewer for this feedback; we have incorporated these changes and improved the writing in identified places to signal this.

---

### Official Review · Reviewer_vsBS · 2025-10-31

**Soundness:** 2
**Presentation:** 3
**Contribution:** 2
**Rating:** 6
**Confidence:** 3

**Summary:**

The paper proposes a framework for analyzing prompt injection attacks on tool-using LLM agents. It characterizes the relevant factors an adversary should consider while crafting such attacks including Target, Stealth, Vector, Budget and Timing. It also proposes timing as a new characteristic of prompt injection attacks. It provides reasonable amount of experimental results to observe some trends for each of the factors.

**Strengths:**

1. The paper takes an important step towards characterizing the relevant factors for the success of prompt injection attacks.

2. The paper attempts to provide experimental results for each of the proposed factors.

3. The paper presents attacks on real-world deployments including Claude Code and Gemini-CLI.

4. The writing is clean and easy to follow.

**Weaknesses:**

1. The experimental results have too much variation across models, it doesn't seem fair to draw any single conclusion from the average behaviors (refer Figure 3). Exploring the possible reasons for such huge fluctuations is an important task to claim anything about the transferability of the conclusions.

2. The paper doesn't discuss about the stealth (one of the factors in framework) concretely in the experiments.

3. The target for all the experiments is again fixed which might raise concerns about the transferability of conclusion (mentioned in the limitations as well).

4. All these factors are not independent, studying the correlation between factors would be important to actually gauge their importance. For example, the experimental results for the timing implies that it is optimal to provide the trigger context and the opportunity to corrupt the response at the same time, is there any benefit in keeping them separate in terms of stealth ?

**Questions:**

1. Is there any analysis/conclusions for each LLM that may provide adaptive attacks and defenses tailored to each model ?

---

> ### Author Response · Authors · 2025-12-03
>
> We thank the reviewer for their detailed feedback. We would like to offer the following responses to points raised.
>
> > The experimental results have too much variation across models, it doesn't seem fair to draw any single conclusion from the average behaviors (refer Figure 3). Exploring the possible reasons for such huge fluctuations is an important task to claim anything about the transferability of the conclusions.​
>
> We would emphasize that the variability of the results is itself a key message of our paper – while some models are more or less resistant against these different types of prompt injection attacks, without testing these currently-untested scenarios, we cannot know for certain how they will respond.
>
> > The paper doesn't discuss about the stealth (one of the factors in framework) concretely in the experiments.​
>
> We note that we detailed our logic for excluding this factor from consideration in Sections 3 and 4 – as other work has considered various settings of the “stealth” parameter, ranging from the agent being allowed to take any action unsupervised, or its chain-of-thought reasoning being actively monitored. Instead, we focused on the novel and underexplored areas we identified.
>
> > All these factors are not independent, studying the correlation between factors would be important to actually gauge their importance. For example, the experimental results for the timing implies that it is optimal to provide the trigger context and the opportunity to corrupt the response at the same time, is there any benefit in keeping them separate in terms of stealth ?​
>
> We note an inherent issue here with the combinatorial complexity of varying each factor with every other factor. We hope that our experiments sampled some key parts of the attack space that provide sufficient evidence to our main points.
>
> > Is there any analysis/conclusions for each LLM that may provide adaptive attacks and
> defenses tailored to each model ?
>
> In production, we expect all attacks and defenses to adapt in a changing landscape. Regarding the conclusions of this work in particular, we could, for instance, select the best settings re: Vector, Budget, Timing, and see how much this improves attack performance, but we expect prompt injections in practice to be largely determined by opportunity – so an attacker will use whatever vector they can manage, with the attack being introduced at a random timestep, and so on.

---

### Official Review · Reviewer_NN78 · 2025-11-01

**Soundness:** 1
**Presentation:** 1
**Contribution:** 2
**Rating:** 2
**Confidence:** 4

**Summary:**

This paper studies long-horizon prompt injection attacks on LLM-based agents, focusing on cases where malicious instructions injected into the context may activate dozens of turns later. The authors propose a five-axis framework (Target, Stealth, Vector, Budget, Timing) to describe prompt injection attacks, conduct controlled experiments on synthetic multi-turn dialogues, and test zero-shot transfer to real deployed systems such as Claude Code and Gemini-CLI. The results suggest that some commercial systems remain vulnerable even after many turns, and that reasoning depth can affect attack success rate.

**Strengths:**

The problem setting of studying long-horizon prompt injections is novel and intuitively important, especially as LLM agents increasingly operate in multi-turn environments.

**Weaknesses:**

- Lack of formalization: The notion of “long-horizon prompt injection” is only loosely defined. The paper does not provide a clear formal threat model, trigger definition, or precise criteria for what constitutes a long-horizon attack versus a normal prompt injection. The attacks resemble backdoor-style conditional activation rather than classical prompt injection, but the paper does not clearly justify this distinction.

- Evaluation methodology is non-standard and underspecified with limited diversity of the evaluation set: Only one attack type (inserting remote code execution vulnerabilities) is studied. The dataset lacks diversity across tasks, modalities, or realistic user contexts.

- Writing and presentation issues: The paper’s structure and language are below top-conference standard.

**Questions:**

N/A

---

> ### Author Response · Authors · 2025-12-03
>
> We thank the reviewer for their detailed feedback. We would like to offer the following responses to points raised.
>
> > Lack of formalization: The notion of “long-horizon prompt injection” is only loosely
> defined. The paper does not provide a clear formal threat model, trigger definition, or
> precise criteria for what constitutes a long-horizon attack versus a normal prompt
> injection. The attacks resemble backdoor-style conditional activation rather than
> classical prompt injection, but the paper does not clearly justify this distinction.​
>
> We agree that we used an informal definition of our new, expanded threat model of prompt injection. We will update the manuscript with a brief formalization inspired by other research in the field.
>
> > Evaluation methodology is non-standard and underspecified with limited diversity of
> the evaluation set: Only one attack type (inserting remote code execution
> vulnerabilities) is studied. The dataset lacks diversity across tasks, modalities, or
> realistic user contexts.​
>
> We agree that this is a limitation of our work, as we discussed in Section 5. However, we expect that this would, at worst, affect the magnitude of our main result, rather than cause a qualitative change in how agents respond to long-horizon prompt injection. To reiterate, we choose a task that is explicitly instructed against in the system prompt; standard prompt injection tasks like uploading user credentials are not explicitly countermanded, so we expect ASRs to be higher in this case.
>
> > Writing and presentation issues: The paper’s structure and language are below
> top-conference standard.​
>
> We note that other reviewers found the paper’s writing to be clear and legible.

---

### Official Review · Reviewer_TZzu · 2025-11-04

**Soundness:** 2
**Presentation:** 3
**Contribution:** 2
**Rating:** 6
**Confidence:** 4

**Summary:**

The authors present empirical analyses of long-horizon prompt injections, those in which the trigger and the attack opportunity are separated by a large number of conversation steps, and find that simple attacks are successful in this context; for certain models, this is true across system prompt, user message, and tool response attack vectors.

**Strengths:**

- The authors present a large number of useful experimental results, which indicate that a large number of commonly deployed language models are vulnerable to long-horizon attacks.
- Results on real-world systems (e.g. Claude Code) are presented, and the examples of malicious behavior are useful.

**Weaknesses:**

- The conclusions drawn from the results by the author are not fully substantiated and generally speculative. While "testing a model's resistance to prompt injection at t=0 may not provide a strong indication.." [307] is intuitive, the results in Figure 3 do not clearly justify this conclusion, and generally show that long-run ASRs fluctuate about the short-run values for the majority of values. Similarly, while strong effects are visible for certain LLMs, the effect of reasoning effort in Figure 4 appears to be weak for the majority of models. It is unclear whether variance impacts these results; the experiments should be conducted over multiple runs and standard hypothesis tests should be performed.
- The results for ASR vs attack budget in Figure 3 actually measure the effects of summarization on the specific prompt injections used, not the adversary's budget, and should be described as such. Performance of a worst-case adversary should never decrease with budget.
- The results are highly variable by model, and hence it is unclear to what exent these results generalize.
- While the results presented here are a useful first analysis of long-horizon attacks, the notion of "long horizon" considered in this work is somewhat limited, and doesn't consider more subtle behavior, e.g. where the desired behavior is to delay defection and behave benignly initially.

**Questions:**

In Figure 4, ASR is, as might be expected, by far the highest when the attack opportunity is immediately following the trigger. How does this vary over a shorter time horizon (e.g. for additional timesteps in the 40-50 range)?

---

> ### Author Response · Authors · 2025-12-03
>
> We thank the reviewer for their detailed feedback. We would like to offer the following responses to points raised.
>
> > The results for ASR vs attack budget in Figure 3 actually measure the effects of
> summarization on the specific prompt injections used, not the adversary's budget, and
> should be described as such. Performance of a worst-case adversary should never
> decrease with budget.
>
> We agree with the reviewer on this point regarding a worst-case adversary. However, practical adversaries are not necessarily worst-case adversaries, and may not be able to iterate against a specific system’s defenses to identify which longer token sequences are more effective than they are suspicious. We will add language clarifying this point.
>
> > The results are highly variable by model, and hence it is unclear to what exent these results generalize.
>
> We would emphasize that the variability of the results is itself a key message of our paper – while some models are more or less resistant against these different types of prompt injection attacks, without testing these currently-untested scenarios, we cannot know for certain how they will respond.
>
> > While the results presented here are a useful first analysis of long-horizon attacks, the
> notion of "long horizon" considered in this work is somewhat limited, and doesn't
> consider more subtle behavior, e.g. where the desired behavior is to delay defection
> and behave benignly initially.
>
> We note that this is exactly the behavior used in this piece – once prompt injected, the model delays defection and behaves benignly, until a trigger condition is seen.

---

### Meta-Review · Area_Chair_P4je · 2026-01-07

**Summary:**

The paper investigates long-horizon prompt injection attacks, where malicious triggers and actions are separated by significant delays. The authors propose a descriptive framework and validate their findings on synthetic dialogues and production systems like Claude Code and Gemini-CLI. While reviewers acknowledged the novelty of the temporal analysis and the effort to test on real-world systems, significant concerns regarding soundness and evaluation remain. Reviewers highlighted that the high variance in empirical results makes the conclusions speculative and difficult to generalize. Furthermore, fundamental issues were raised regarding the threat model (focusing on direct system prompt modification rather than indirect injection), the lack of formal definitions for the proposed framework, and the limited evaluation scope which relied solely on Attack Success Rate without assessing the impact on benign task utility.

**Reviewer Concerns:**

Addressed by Rebuttal:

- Threat Model Relevance: The authors clarified that for coding agents, modifying the system prompt is a realistic threat model, which addresses some of the reviewer's concerns about practicality. Yet still, the threat model is strong and may not apply to general agent cases

- Clarification of Variability: The authors argued that the high variance in results is a finding in itself (i.e., model behavior is unpredictable), rather than an experimental flaw.

- Formalization: The authors promised to include formal definitions for their threat model and "long-horizon" attacks in the camera-ready version

Outstanding:

- Soundness of Conclusions: The high variance in results remains a critical issue. The rebuttal did not fundamentally resolve the concern that the data may be too noisy to support the paper's claims.

- Evaluation Metrics: regarding the lack of utility benchmarks remains largely unaddressed practically. The authors argued that benign performance was ~100%, but did not provide rigorous data to support this trade-off analysis.

- Practicality of Delayed Attacks: Reviewers questioned whether a delayed attack is practically more dangerous or distinct enough from standard backdoor/conditional attacks to warrant a new category, a conceptual gap that remains.

**Reviewer Scores:**

likely to remain unchanged

---

### Decision · Program_Chairs · 2026-01-26

Reject